# Going Forward from B to A? Proposals for the Eurozone Crisis

**Massimo Amato** [1,†]**, Luca Fantacci** [1,†]**, Dimitri B. Papadimitriou** [2,†] **and Gennaro Zezza** [3,*,†]

[1]   Department of Policy Analysis and Public Management, Università Bocconi, Milano 20136, Italy;
      massimo.amato@unibocconi.it (M.A.); luca.fantacci@unibocconi.it (L.F.)
[2]   Levy Economics Institute, Bard College, Annandale-on-Hudson 12504, NY, USA; dbp@levy.org
[3]   Department of Economics and Law, Università di Cassino e del Lazio Meridionale, Cassino 03043, Italy
[*]   Correspondence: zezza@unicas.it; Tel.: +39-0776-2994641
[†]   Authors contributed equally to this work.

**Abstract:** After reviewing the main determinants of the current Eurozone crisis, this paper discusses the feasibility of introducing fiscal currencies as a way to restore fiscal space in peripheral countries, such as Greece, which have so far adopted austerity measures in order to abide by their commitments with Eurozone institutions and the IMF. We show that the introduction of fiscal currencies would speed up the recovery, without violating the rules of Eurozone Treaties. At the same time, these processes could help the transition of the euro from its current status of single currency to a status of "common clearing currency" along the lines proposed by Keynes at Bretton Woods as a system of international settlements. Eurozone countries could therefore move from "Plan B" aimed at addressing member state domestic problems, to a "Plan A" of a better European monetary system.

**Keywords:** euro; fiscal currencies; austerity; current account imbalances, Clearing Union

## 1. Introduction

An increasing number of economists and commentators believe that the current (Spring 2016) economic policy path that some Eurozone countries are following will undermine the rules of the European Monetary Union (EMU) originally put in place in the Maastricht Treaty in 1992, and subsequently modified in 2007 by the Lisbon Treaty, and in 2011 with the "Sixpack", and will eventually lead to either the collapse of the European cohesion or a period of prolonged stagnation.

The rules of the EMU structure were based on two assumptions, both of which have proven to be untenable. The first was the belief in a smooth transition from simple agreements among different national States to a federation, creating the "United States of Europe"[1], which would complete the institution of a common market, but also share the same Constitution, thereby ensuring common rights for the "European citizen", a common foreign policy, an integrated fiscal system and a common currency. Had this belief been realized and accomplished smoothly with the approval of the population of each member state, then the currently missing institutional mechanism of a unified fiscal structure sufficiently large enough to be an automatic stabilizer facilitating federal fiscal transfers to member states at times of need would have made the Eurozone sustainable. This process was, however, stopped

---

[1]   The "Ventotene Manifesto" is believed to be one of the major sources of inspiration for a plan towards a federation of European countries. See [1].

by the ill-conceived proposal for a European Constitution, which, albeit ratified by several member states, was rejected by the French and Dutch voters in 2005, de facto halting any further attempts to put the United States of Europe project on strong foundations.

The second assumption inspiring the logic of the Maastricht Treaty, and its subsequent modifications, was based on the ordo-liberal economic dogma that prevailed then and continues to this day mainly by Germany's dictum. It asserts that markets will self-adjust towards full employment, the Central Bank should be independent from governments and be concerned only with price stability, and national governments should be responsible for fiscal policy subject to the Treaties' guidelines, guarantee property rights and smooth the functioning of markets irrespective of the asymmetries in their real economies. This (Bundensbank) logic inspired the structure of the ECB, avoiding the possibility of acting as lender of last resort to governments, if needed; it also inspired the limits to government deficits and debts codified in the Maastricht Treaty, and made even more stringent in the Six Pack.

In the public debate that ended with the signing of the Treaties, it became clear that the adoption of a single currency will mean the renunciation of domestic authorities in having any role in the formulation of monetary and exchange rate policies. This, on the face of asymmetric shocks, would imply divergence and crisis handling among member States in accordance with their underlying real economies. To spur growth in regions lagging behind, a system of fiscal transfers—the Structural Funds and the Cohesion Fund—was therefore established. Moreover, some provisions were later included in the Treaties to force countries whenever their current account balance exceeded a given threshold, relative to their GDP, to take corrective actions reversing their surplus positions. The mechanism of fiscal transfers, however, is insufficiently funded to act as an automatic stabilizer at the level experienced in the United States with a sufficiently large federal budget in the order of 15 percent of GDP while the requirement for surplus-reversing—introduced only in 2011 as part of the "Macroeconomic Imbalance Procedure"—has not been applied thus far, despite the large external surpluses of Germany (more than eight percent of its GDP) and other countries of the European North.

The euro's faulty architecture was thus well known before its implementation started [2,3][2]. In a prescient contribution, Godley wrote:

> " . . . if all these functions are renounced by individual governments they simply have to be taken on by some other authority. The incredible lacuna in the Maastricht programme is that, while it contains a blueprint for the establishment and modus operandi of an independent central bank, there is no blueprint whatever of the analogue, in Community terms, of a central government" [5].

The paper is structured as follows: In the next section, we briefly review the historical evolution of the imbalances which led to the prolonged stagnation of the Eurozone periphery and the outright crisis dominated in Greece—which have stimulated a debate on whether to reform the Eurozone institutions (a plan commonly referred to as "Plan A"), or focus instead on implementing domestic policies that do not necessarily terminate the current Eurozone agreements ("Plan B")[3]. There is also the option that countries in deep crisis, such as Greece, may very well choose to abandon the euro, possibly precipitating the collapse of the Eurozone monetary system. In this paper, we will not investigate this possibility. Instead, in the third section, we present a proposal, based on [6–9], for the introduction of a domestic fiscal currency compatible with keeping the euro as legal currency. In the fourth section, we discuss how some simple changes in the functioning of the ECB Target2 system may lead the way to a more sustainable monetary architecture and, in the final section, we offer our conclusions.

---

[2]　On the other hand, there was a hope that countries not satisfying the requirements for an Optimal Currency Area would converge to such requirements once the common currency had been adopted. See [4] among others.

[3]　The use of the terms "Plan A" and "Plan B" in this context possibly started from Varoufakis's proposals for an alternative to austerity in Greece, and have been used in the political debate. See for instance www.euro-planb.eu.

## 2. The Rise and Fall of the Eurozone Periphery

The adoption of the euro as a common currency afforded some benefits to the Eurozone (EZ) member states of the periphery, but concurrently, its faulty design planted the seeds of the current crisis and exacerbated the already unsustainable trajectories.

The adoption of the single currency, accompanied with a single monetary policy, had two major implications:

(a) Domestic banks could obtain reserves from the ECB at the same discount rate. This meant a significant reduction of interest rates in the EZ member states of the periphery (Figure 1), reducing the cost of borrowing to firms and households, converging with the already low interest rates of the core EZ member states, such as Germany. As Figure 1 shows, the decline in interest rates was similar as in the United States, a consequence of the decline in inflation worldwide;

(b) The single currency eliminated the possibility of exchange rate adjustments. The value of the euro against the US dollar, or other currencies depended on the overall price competitiveness of the area as well as on financial markets. As shown also in Figure 1, however, long-term interest rates in the EZ converged to those in the United States when the euro was introduced, so that financial speculation on exchange rate markets did not appear to have played a major role during 2000–2007. This implied that even small differences in inflation among EZ countries built up on relative price competitiveness, made the euro a strong currency for countries with higher inflation above the EZ average, such as Italy or Greece, and a weak currency for countries with inflation below the EZ average, such as Germany. To phrase the process differently, Germany and other low inflation countries would have seen their exchange rate appreciate, had they kept their domestic currency, while Italy, Spain or Greece would have depreciated their domestic currency, as long as their inflation rates remained above those of Germany.

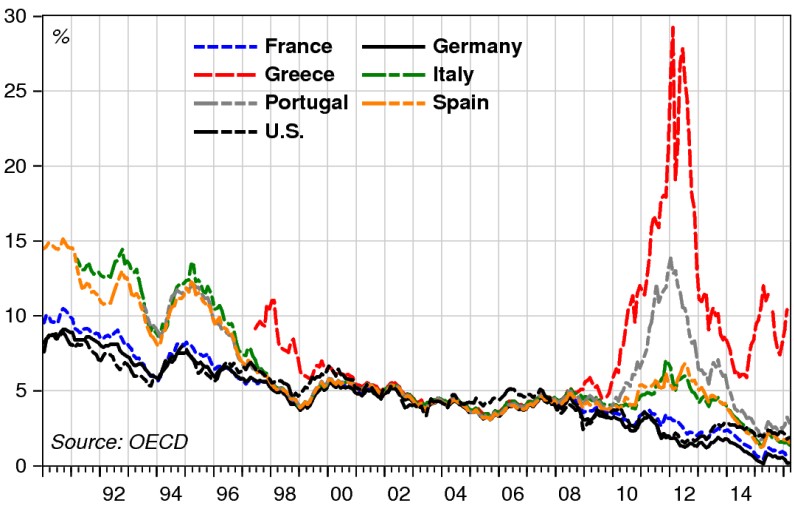

**Figure 1.** Long-term interest rates.

The former process allowed some countries in the EZ periphery—notably Greece, Ireland and Spain, to achieve higher rates of growth in their real per-capita GDP, in the 2000–2007 period, closing the gap with Germany (Figure 2). In some cases, such growth rates undoubtedly were achieved through private sector borrowing, helping generate asset inflation in the housing sector: this was the case for Ireland, Spain and—to a lesser extent—Greece. In the same 2000–2006 period, Germany was considered to be the "sick man of Europe", experiencing low growth rates, and while the introduction of labor market reforms helped lower the unemployment rate, contribute to a strong wage moderation [10], and augment price competitiveness, they failed to stimulate domestic demand.

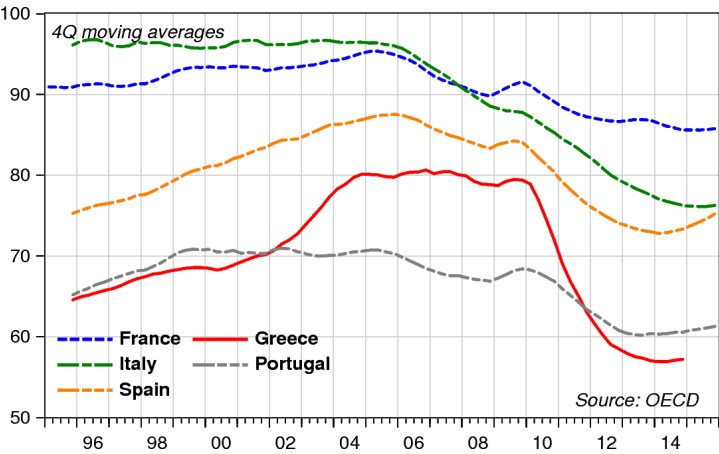

**Figure 2.** GDP per capita (Germany = 100).

The combined processes—faster growth and reduced price competitiveness in the periphery relative to the core—had a strong impact on current account balances relative to GDP, which turned negative for surplus countries, like France and Italy, and deteriorated further in deficit ridden countries, such as Greece and Portugal (Figure 3). Greece and Portugal, in particular, adopted the euro with a large current account deficit relative to GDP, leading to an unsustainable future increase in their net foreign debt position. The impact on their trade of relatively faster growth and decreasing price competitiveness made matters worse.

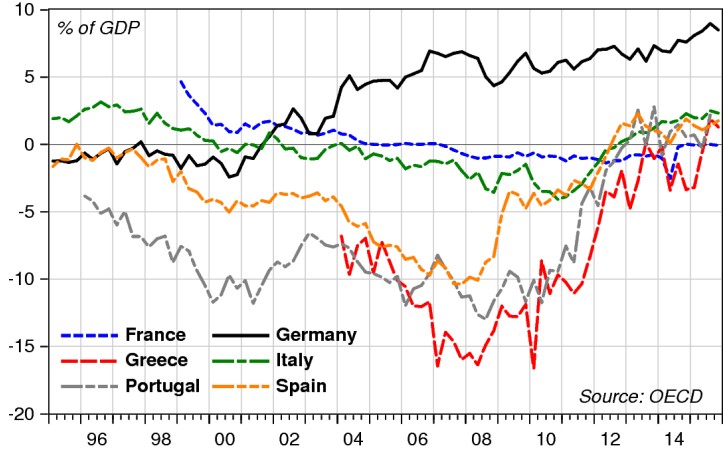

**Figure 3.** Current account balances.

The growing trade imbalances were and still are even more evident when we focus on the trade of the periphery with the core. In Figure 4a–e, we report bilateral trade in goods, scaled to the GDP of Germany. The adoption of the euro turned the balance of trade for France, Italy and Portugal into a trade deficit, and increased further the trade deficit for Spain and Greece. Such trade imbalances were obviously funded via an accumulation of private debt in the periphery mirrored as credit in the core countries. As long as countries kept growing, however, such debts were not considered a major problem, especially since the financial markets were expecting the ECB to intervene, should a crisis happen, even against its mandate.

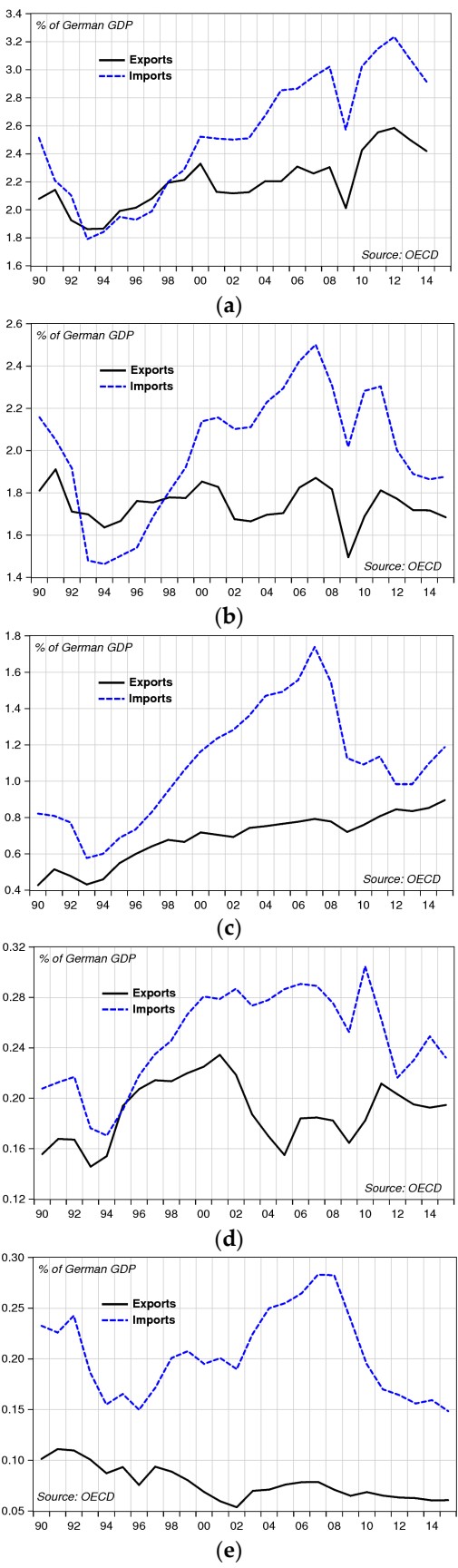

**Figure 4.** Bilateral trade in goods with Germany. (**a**) France; (**b**) Italy; (**c**) Spain; (**d**) Portugal; (**e**) Greece.

When the Great Recession hit in 2007–2008, popping the bubbles in the housing markets in the United States and Spain, Ireland and Greece, it disrupted the stability of European banks' balance sheets loaded with toxic US financial assets. This revealed the fragility of growth in many EZ countries that required governments to come to the rescue of a collapsing financial system. At the same time, the recession lowered the actual and prospective income of the over-borrowed households and firms, negatively affecting domestic demand with implications of higher unemployment. In these countries the private sector started to deleverage, and many loans became non-performing, putting further stress on banks' balance sheets. In all EZ countries, the recession inflicted the standard impact on public finances, i.e., a reduction in tax receipts, and larger welfare payments in the face of rising unemployment. In some countries, i.e., Ireland, the government chose to bailout banks in trouble, transforming large portions of private debt into public debt.

In Greece, when the George Papandreou government took office in 2009, both the public deficit and debt were higher than reported by the previous government, triggering an adverse reaction from the EZ institutions, and concurrently the financial markets, precipitating the country's sovereign debt crisis. An obvious option for Papandreou would have been to either insist for a debt write-down supported by the IMF or to abandon the euro and redenominate all debt obligations under Greek law in a new national currency. This possible option, along with the possibility of an outright default—understood by the players in the financial markets—caused prices of Greek bonds to collapse precipitously and interest rates to rise to unprecedented levels (Figure 1). Papandreou chose instead to accept an agreement with the Eurozone institutions and the IMF, to obtain new liquidity—albeit, as it turned out, at an insufficient level—on the condition of implementing fiscal austerity, and structural labor market reforms including severe wage and pension cuts.

The outcome of the Greek crisis gave a clear signal to financial markets that the ECB would not act as lender of last resort should a member state of the EZ run into peril. The consequence of this was the recognition that the euro had the status of a foreign currency, when repaying debt held by non-residents[4]. The other implication was that governments in the EZ, having lost their monetary sovereignty, could default, and this prospect generated the large spread in interest rates of other EZ countries, as reported in Figure 1.

The implementation of fiscal austerity was backed by theories suggesting that the fiscal multiplier was small, or even negative[5], so that reductions in public expenditure—believed to be more effective than increases in taxation—would not result in large drops in GDP. After a few years of austerity in the EZ periphery countries, fiscal multipliers proved to be much larger, especially during a recession, to the consternation of the IMF chief economists and others who willingly or unwillingly admitted it [14]. Notwithstanding the discrediting of the "expansive austerity" theory, fiscal austerity and labor market reforms are still the TINA (There Is No Alternative) recommendation for countries like Greece: in currently ongoing negotiations (April 2016) the IMF is demanding the implementation of further cuts in pension payments and other public expenses including precautionary austerity should the Greek government fail to meet the agreed upon primary budget surplus 2–3 years from now.

Once it is recognized that the problem of the Eurozone periphery is not the size of public debts and deficits per se, but rather the size of private and public debt held abroad (on the asset side of the balance sheet of financial institutions in creditor countries), fiscal austerity, together with labor market reforms that cut nominal wages substantially, is thought to be an effective short-term instrument to make such debt sustainable, thereby increasing price competitiveness. Theory suggests that, in concert with this comes increased profitability; thus making the policy of austerity an effective—albeit painful—long-term solution.

---

4    See [11] for a discussion.
5    See [12] for a discussion, and [13] for a meta-analysis of estimates of multipliers.

Fiscal austerity, through its strong effects on domestic demand, also reduces imports that helps improve the trade balance in the short-term and guarantees a net influx of euro reserves to service foreign debt—even at the cost of rising unemployment and a private sector in misery.

Theory does suggest that the fall in nominal wages may translate into increased price competitiveness which, effective on trade, may reverse a current account deficit into a surplus. It also suggests that labor market reforms with the potential effect of increased profitability will, in turn, translate into higher investment, stimulating economic growth.

Looking at the experience of Greece, where nominal wages have fallen by 22 percent between 2009 and 2015, and by 27 percent in real terms, only part of the fall in wages has translated into lower prices, and lower prices have improved trade only for the tourism sector. At the same time, lower real wages have contributed, along with fiscal austerity, to keeping domestic demand very low, and thus increasing uncertainty over the future profitability of private investment.

The effectiveness of austerity as a tool to achieve growth through increases in net exports is obviously diminished when all countries in a trade agreement—as it is the case for the Eurozone—pursue the same strategy. If wages and prices are falling among all trade partners, nobody is getting a relative advantage. This constitutes a race to the bottom that nobody wins [2,3]. It is not surprising, therefore, that the improvement in trade of many EZ countries is with trade partners outside the Eurozone.

At the time of writing, it seems that fiscal austerity and more labor market reform will continue to be the only policy option administered by Eurozone institutions (and the IMF) to the EZ periphery countries. We believe that, if this continues, the improvement in current accounts will not be sufficient to offset the negative impact of austerity, and the area will experience modest growth rates or stagnation for many years to come, unless the balance sheets of the entire EZ financial sector are very significantly strengthened.

In the face of the possibility of prolonged stagnation, the major concern of EZ institutions seems to have been the strengthening of the asset side of the balance sheet of the EZ banks, and the introduction of alternative monetary policy (Quantitative Easing or QE), increasing the commitment of individual countries to the euro.

The evidence of the above can be drawn from the plan to "rescue Greece" as it has been implemented. As documented in [8], the sum of 200 billion euros in loans, made to the Greek government between 2010 and 2014, was used to decrease Greek public debt held abroad by private investors by 104 billion euros, to support (recapitalize) the Greek financial sector by about 60 billion euros, and to pay interest on outstanding debt of about 52 billion euros. While the EZ media have presented the support to Greece as a transfer from the Eurozone taxpayer to the lazy Greeks, reality has been quite different. There have been no net fund transfers[6] for purposes of employment growth or to stimulate investment. While the Greek public debt outstanding in 2010 had mainly been issued under domestic legislation—and could therefore be redenominated in case of a euro-exit, almost all of it has now been re-issued under foreign law that renders such an option impossible.

Of the clauses of the QE program that the ECB has been implementing since 2015, and more particularly as detailed in the Treaty establishing the European Stability Mechanism for long-term bonds issued after 2013, the covenant of the Collective Action Clause now allows a minority of bond holders to veto an offer from the issuer to change the terms of the contract (as it would be the case if a country chose to exit the euro and redenominate its existing debt in a new currency).

---

[6] We are not denying that European funds have been used for employment programs by the Greek government but these are not included in the rescue packages since the financial obligations more than exceeded the liquidity made available by the "rescue" packages.

More importantly, the QE program started in 2015 has introduced risk sharing clauses entailing that—should a country exit the euro or default on its debt obligations—the full value in euro of such debt would be guaranteed by its domestic central bank. Thus the public debt of countries like Italy, for instance, that has been issued mainly under domestic legislation, and could therefore be redenominated, will now have to be extinguished in a currency, the euro, which will become a foreign currency should those countries choose to exit the EZ.

## 3. Fiscal Currencies to Reflate the Eurozone Periphery

As discussed above, the major concern of the EZ institutions seems to be strengthening the asset side of the balance sheet of EZ banks, and this in turn requires each EZ country to be able to honor its debt—both private and public—in euro. From a macroeconomic perspective, this goal requires countries to run a current account surplus. Any alternative domestic policy compatible with Eurozone Treaties—so called "Plan B" policies—therefore, must comply with this requirement to be politically feasible.

Our proposal, presented in [7,9] for Greece, is based on introducing a fiscal currency, which for Greece we labelled "geuro" following Mayer [15]. It would be issued in a similar fashion as the Swiss WIR [6], and would neither become the new domestic legal tender nor be used to issue geuro bonds to replace existing euro denominated bonds, but be accepted at par strictly for internal transactions and tax payments to central and local governments[7]. This will render the geuro legally and politically compatible with the existing EMU rules.

Several proposals[8] have been advanced for introducing an alternative currency in Greece, with many differences, mainly relating to the convertibility of the new currency into euros, and how the new currency should be issued. We propose that the geuro would be issued in electronic form—not prohibited by the EMU framework—to increase government expenditure for direct job creation and for additional transfers to partially recover the cuts to pensions implemented in Greece, or to fund programs alleviating poverty. The geuro would be accepted for all tax payments, up to a limit of 20 percent, while the remaining tax obligations would be paid in euro. The government would not guarantee the convertibility of the geuro, but since it would be ready to accept it as tax payments at par with the euro, this should sustain the value of the new currency, as long as the amount of new currency in circulation was not "excessive". As such, the geuro would quickly become a close substitute of the euro in settling internal payments among private agents, and there is no obvious reason for prices in geuro to differ from prices in euro, as long as the government keeps the credibility in the program. An independent institution would be established to monitor the supply of geuro and control any inflationary pressures ensuring the success of this alternative plan.

The amount of geuro to be issued would be fixed subject to the following two constraints:

1.  The increase in domestic demand, and therefore imports, that has to be expected from this fiscal stimulus should not generate a current account deficit, and,
2.  The relation between geuro issued through additional expenditure and geuro retrieved as tax receipts should not compromise the government targets on the fiscal deficit in euro, which have been established in the negotiations with the Eurozone institutions and the IMF.

As we have shown elsewhere [7–9], our estimates of the impact of such proposals based on simulations of the Levy Institute's Model for Greece (LIMG) show that a moderate fiscal expansion, financed by issuing geuro, would increase the growth rate in real GDP considerably, while keeping the current account in surplus. Given the rising poverty, and high unemployment rate—hovering around

---

[7]   We will not attempt here a survey of the literature on parallel currencies. For a proposed taxonomy, see [16]. See [17] for an interesting analysis on how to reintroduce domestic currencies while keeping the euro as a parallel currency. See [18] for a different proposal of a fiscal currency.

[8]   We will not discuss alternative proposals here. See our survey in [7]. See also [19,20].

24 percent—the newly employed labor, together with low-income pensioners paid partially in geuro, would expend their geuro for purchasing domestically produced goods, as opposed to those imported which require payment in euro, and in so doing would foster an increase in the production of domestic goods, increasing GDP while still maintaining the current account surplus. Thus, the demand for euro will not increase, enabling the government to keep a surplus in euro, while generating a deficit in geuro. It is expected that the amount of additional expenditure in geuro would exceed tax receipts in geuro, at least for the first years of the program to stimulate growth.

There is no reason why the introduction of a fiscal currency, which is not considered legal tender, should be opposed by the current Eurozone partners under the Treaties, as long as it does not compromise existing obligations which, as we have argued, are mainly related to the ability of governments to meet the deficit and debt targets in euro under the agreements detailed in the Memoranda of Understanding.

This proposal, however, should not be limited to Greece. The introduction of fiscal currencies in other countries, like Italy, would allow them to gain back the needed fiscal space, and get their economies out of stagnation[9].

## 4. The Euro as a Common Currency?

As recalled above, a necessary condition for a "Plan B" to be feasible is that it helps a country meet its public and private commitments in euro. In other terms, the success and the acceptability of a "Plan B" on the part of Eurozone institutions depends on its prospective capacity to contribute to a reduction of both the public and foreign debt of the country involved.

The introduction of a fiscal currency along the lines illustrated in the previous section, by reducing the budget deficit under the assumption of a multiplier effect, could provide a positive contribution towards the attainment of the first objective. However, it would also run the risk of leading away from the second objective, by possibly amplifying the trade deficit. The fiscal currency, by expanding aggregate demand, could, in fact, produce the undesired side effect of weakening the external position of the country through two different channels. First, to the extent that the increase in aggregate demand should lead, directly or indirectly, to an increase in the demand for imported goods, the budget deficit financed by the fiscal currency would produce a proportionate expansion of the trade deficit. Second, by alleviating deflationary pressures, the expansion of domestic demand would affect the terms of trade, mitigating and possibly offsetting the competitiveness gains pursued by means of austerity measures.

The proposal of a reformed Target2 system sketched out in the present section[10] aims expressly at reducing balance-of-payments disequilibria within the Eurozone, and is intended therefore, in this respect, as an ideal complement to the fiscal currency outlined above. Just as the latter addresses the reduction of public debts in euro, the proposal presented here tackles foreign debts by overturning the logic that inspires austerity policies: instead of impinging exclusively on debtor countries to have them reduce their imports, inducing a contraction of intra-European trade and a mercantilist stance of the Eurozone towards the rest of the world, it aims at producing an expansionary effect by also involving creditor countries and encouraging them to import more.

The idea that "creditors should not be allowed to remain passive" was at the basis of Keynes's proposal to reform the international monetary system after World War II. In his view, a surplus country ought to be induced to dispose of its credit, in order to facilitate the repayment of debts by deficit countries and to avoid exacerbating deflationary pressures on international trade, which would eventually damage, by repercussion, the creditor countries as well.

---

[9]  Similar proposals are being discussed in Italy, based on the government issuing Fiscal Credit Certificates, rather than a fiscal currency proper. We evaluated this proposal for Greece in [8]. See [21] for a recent view.

[10]  See also [22].



To apply the principle of symmetrical adjustment, Keynes's plan envisaged the establishment of an International Clearing Union, which would act as a bank for the settlement of all payments related to international trade, and would finance temporary imbalances simply by crediting the account of the exporting country and debiting the account of the importing country. However, unlike a regular bank, the Clearing Union would charge a fee not only on debtor countries, but also on creditor countries, since both would be benefiting from the services of the Clearing Union: the deficit countries, by allowing them temporarily to buy more than they would have otherwise afforded to buy; but also the surplus countries, by allowing them to sell more than they would otherwise have been able to sell. Moreover, the symmetric charges would act as an incentive for all countries to converge towards a balanced trade[11].

It is interesting to note that Keynes's plan explicitly excluded the adoption of austerity measures as a means to restoring external equilibrium:

> "the measures [ . . . ] which the Governing board [of the Clearing Union] can ask a country to take 'to improve its position', if it has a substantial debit balance, do not include a deflationary policy, enforced by dear money or similar measures, having the effect of causing unemployment; for this would amount to restoring, subject to insufficient safeguards, the evils of the old automatic gold standard" [23], p. 143.

The principles underlying Keynes's plan could clearly provide an alternative approach to the balance-of-payments disequilibria within the Eurozone described in Section 2. Their practical implementation could rely on a clearing system, Target2, which already exists and is managed by the ECB for the settlement of intra-Eurozone financial transactions. Since the outbreak of the current crisis, Target2 has played a crucial role in financing balance-of-payments disequilibria within the Eurozone by creating a form of reserve asset similar to Bancor. However, unlike the Clearing Union, it is used to finance not only trade deficits, but also, and primarily, capital flights [24]. Moreover, it fails to reabsorb disequilibria given that Target2 balances, unlike Bancor balances, are not subject to quotas or to symmetrical charges on surplus and deficit countries. In fact, only negative balances are currently charged with the payment of interests to the ECB, while positive balances are "generally remunerated at the respective interest rate for the main refinancing operations" [25][12].

An outright transformation of Target2 into a European Clearing Union built upon the blueprint of the Keynes Plan (as envisaged in [27]) would entail major changes that do not appear to be politically viable, such as the adoption of stringent capital controls, the reintroduction of national currencies, and the setting of quotas on the balances of individual member states. A more modest, yet more realistic proposal, would be simply to introduce symmetric and increasing charges on positive and negative Target2 balances. A similar measure would induce surplus countries to play their part in restoring balance-of-payments equilibria by adopting expansionary policies (such as tax cuts or wage increases) in an effort to restrain exports and encourage imports.

To be sure, under free capital movements, a creditor country need not run a trade deficit in order to reduce its Target2 surplus: it may simply increase its capital lending abroad, particularly through the interbank market. If negative interest rates on excess reserves and Target2 accounts were successful in inducing financial intermediaries to increase lending from surplus to deficit countries, this would be equally important in reducing imbalances across the Eurozone (provided that the loans were actually

---

[11] Keynes's proposal was not adopted at the Bretton Woods conference that defined the post-war global economic order in 1944. Only a few years later, however, it was taken as a model for the creation of the European Payments Union. From 1950 and 1958, the EPU intermediated 75 percent of commercial transactions between member countries, contributing significantly to the "economic miracles" of the time, like Italy and Germany, and to the beginning of the process of European integration.

[12] In fact, when the ECB introduced a negative deposit facility interest rate on 11 June 2014, the latter was supposed to be extended also to account balances in Target2 [26]. However, upon explicit request of the authors, the European Central Bank confirmed that "The remuneration of Target2 positions is calculated daily at the latest available marginal interest rate used by the Eurosystem in its tenders for main refinancing operations. As the interest rate mentioned before is currently at 0% the remuneration of Target2 positions is currently at 0%. This applies for both claims and liabilities".

and responsibly employed to enhance capital development in debtor countries). Therefore, the fact of not introducing the capital controls envisaged by Keynes's original proposal may not prove to be a drawback, but indeed a positive contribution to the reduction of the financial fragmentation of the Eurozone.

Moreover, the proceeds of the symmetric charges on positive and negative Target2 balances would generate a substantial return for the ECB, which could be used to finance investments possibly through the EIB (European Investment Bank) or the EIF (European Investment Fund). The possibility of introducing such charges should fall entirely under the current capacities of the ECB, without representing a form of fiscal transfer between member states and without infringing other EU regulations.

In fact, since the introduction of the Macroeconomic Imbalance Procedure (MIP) in 2011, even the European Monetary Union has started to formally recognize excessive surpluses, alongside deficits, as a source of potential instability. Yet the acknowledgement of the creditors' responsibility falls short of establishing an exact symmetry: deficits are sanctioned when they exceed 4 percent of GDP, while surpluses are allowed to reach 6 percent of GDP without being considered excessive. The asymmetry possibly reflects the fact that, for the purpose of the MIP, deficits and surpluses are measured in terms of the current account of each member state with the rest of the world: to the extent that surpluses reflect net exports outside the Eurozone, they should be regarded not as a factor of imbalance, but of strength for the Monetary Union. Hence, perhaps, the greater indulgence of the MIP for surpluses. However, it would be more sensible to actually distinguish what part of the balance of payments may actually be regarded as a potential source of instability: in this perspective, Target2 imbalances would provide a more appropriate gauge, since they measure the cumulative effect of both current and capital account imbalances between each member state and the rest of the Union. If it were not politically or juridically viable to introduce symmetric charges on Target2 imbalances at the ECB, the reabsorption of balance-of-payments disequilibria within the Eurozone could be ensured by defining surpluses and deficits in terms of Target2 balances and introducing symmetric thresholds and sanctions for excessive imbalances in both directions through the MIP.

The introduction of a similar cooperative mechanism of adjustment would reduce the financial imbalances, which enhance instability and speculation, and give a decisive impetus to the actual circulation of money in the circuits of the real economy. A balanced trade would thus actually operate as a cohesive force for the unification of Europe. The proposed change would not require, but could prepare, a more radical reform of the architecture of the monetary union.

Even if it would not attain the ideal objective of a European Clearing Union, the introduction of a principle of symmetry between creditors and debtors within the Eurozone would imply a complementarity between domestic reflation and external balance, offering the compromise of a pragmatic solution consistent with current European Treaties. It is foreseeable, and not merely desirable, that the positive effects of a similar solution would exceed the difficulties of implementation, thus representing an important precedent for the revision of the Treaties and for a more radical reform of the European monetary architecture, in the direction of a complementarity between the common currency used as a unit of account within a European clearing union and a multiplicity of national, or even regional, currencies linked by a system of adjustable pegs.

## 5. Conclusions

In this paper we have argued that the current rules governing the Eurozone are the result of a faulty architecture and an incomplete political process towards greater European integration that, should it have been completed, would have instituted a mechanism of automatic stabilizers being channeled through a European Treasury with a sufficient budget, as is the case with the United States.

However, be that as it may, we have argued that current account imbalances that were already large for some countries when the euro was introduced, were exacerbated by the adoption of the common monetary policy. The remedy took the form of uncoordinated wage policies, and public investment cuts, leading to a real appreciation of the exchange rate in the Eurozone periphery countries and a real depreciation for countries in the core.

Deficit countries, like Greece, Portugal, and—to a lesser extent—Spain and Italy, saw their net foreign asset position deteriorate, and the need to rescue a collapsing financial system—when the Global Financial Crisis turned into the Great Recession in 2007–2008—contributed to the increase in fiscal deficits and debts that are the consequence of the crisis, rather than the primary cause.

The dangerous policy of austerity has been imposed harshly on countries seeking financial support, such as Greece, and while the IMF's assessments erroneously projected a modest impact on output and employment, the policy instead delivered completely the opposite outcomes. Notwithstanding the abysmal results of lost output and employment, the same policy prescription of more austerity continues unabated. To be sure, austerity has succeeded in addressing at least in the short-term, the main problem of countries in the Eurozone periphery, namely their current account deficits, by generating a large decrease in imports, and a modest increase in exports. A current account surplus is the pre-condition for a country to be able to pay its foreign debt in euro, which is de facto a foreign currency. The continuing austerity solution, however, is likely to generate more stagnation and long-term unemployment, and prompt, sooner or later, a breakup of the EMU which, if it happens disorderly, will most likely cause another and larger global financial and economic crisis.

To avoid a disorderly collapse of the Eurozone, we have proposed two actions which should be politically, as well as economically, feasible. Countries that need to increase their fiscal space for employment, and investment growth, like Greece, can introduce a fiscal currency, which is not legal tender, but is used to finance a fiscal stimulus and direct job creation while it can be accepted, at par, for tax payments obligations. The size of such stimulus should be calibrated in the manner that it does not cause the current account balance to move in negative territory, enabling countries to keep servicing their foreign debt in euro. Our estimates for the Greek economy show that such policy is feasible, and would be successful in restoring growth and employment. Such fiscal currencies could be issued by all Eurozone countries—and even at regional level—to restore fiscal space without breaking the Treaties, while a much needed reform of the Eurozone monetary system begins to occur.

Moreover, we also suggest a simple measure to restore symmetry in response to Eurozone countries with current account imbalances: adopting one of the features of Keynes's Bancor proposal. We propose to charge interest rates on Target2 surplus balances, so as to create incentives for countries with large surpluses to expand domestic demand, thus contributing to the recovery of the entire Eurozone region.

While undoubtedly the latter proposal will not be supported by countries like Germany—currently running large external surpluses and huge Target2 positive balances to the tune of more than 600 billion euros—our former proposal of introducing fiscal currencies should be backed by countries in the periphery that desperately need to reflate, and from conservative political groups in the core alike, fearing that an unresolved and continuing debt crisis may lead to never-ending transfers of liquidity, burdening the national budgets of the countries in peril.

**Acknowledgments:** The authors wish to thank the participants to a seminar in Paris XIII and two anonymous referees for helpful comments on a previous draft. Any remaining errors are our responsibility.

**Author Contributions:** The paper is a joint contribution by all co-authors.

**Conflicts of Interest:** The authors declare no conflict of interest.

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
