# Peer review of "Going Forward from B to A? Proposals for the Eurozone Crisis"

_economies, doi:10.3390/economies4030018_

Reviewer 1 Report

In the introduction the authors should state more clearly the debate around Plan A and Plan B and should contextualise there contribution in the existing literature.

The  content of footnote 3 should be moved to the text, since it is important for the authors' claim regarding the determination of exchange rates.

The sentence on p. 6, line 162 does not make much sense to me. Please check.

On p. 8, lines 236 ff. the authors claim that fiscal expansion in Greece would have considerable growth effects but would keep the current account in surplus, referring to simulations run with the Levy Institute's model. This needs further elaboration and support. How is this to square with the claim being made on p. 6, line 162, that Greece is amongst the countries with low space for import substitution?

On p. 9, lines 303 ff., the authors claim that Target2 should be further developed and be amended by symmetric charges on excessive balances in order to contribute to current account  rebalanicing. But what about capital flows in this context? Keynes's blueprint for a Clearing Union on which this suggestion is built included capital controls.

Author Response

In the introduction the authors should state more clearly the debate around Plan A and Plan B and should contextualise there contribution in the existing literature.

We have added a reference in new footnote #3 to a recent conference on “Plan B for Europe”. A more complete coverage of the existing, vast literature would be beyond the scope of the paper, which focuses on policy proposals

The  content of footnote 3 should be moved to the text, since it is important for the authors' claim regarding the determination of exchange rates.

We followed the referee’s suggestion

The sentence on p. 6, line 162 does not make much sense to me. Please check.

We have modified the following paragraph to make our argument more clear

On p. 8, lines 236 ff. the authors claim that fiscal expansion in Greece would have considerable growth effects but would keep the current account in surplus, referring to simulations run with the Levy Institute's model. This needs further elaboration and support. How is this to square with the claim being made on p. 6, line 162, that Greece is amongst the countries with low space for import substitution?

Our proposal does not rely on import substitution. We are suggesting that the size of a fiscal expansion should be determined by checking its impact on the current account of the country, i.e. it should not be too large to generate a current account deficit.

In any case, we have further elaborated our point as suggested by the referee

On p. 9, lines 303 ff., the authors claim that Target2 should be further developed and be amended by symmetric charges on excessive balances in order to contribute to current account  rebalanicing. But what about capital flows in this context? Keynes's blueprint for a Clearing Union on which this suggestion is built included capital controls.

We have modified this section to extend the treatment of capital controls, and take care of the referee’ suggestion

Reviewer 2 Report

This is a highly topical and relevant paper that should be published. The authors may want to provide some more information as to why their solution is legally and politically feasible in the current EMU framework. Also some technical details and some more information about the risks of the strategy - what happens if too many Geuros come into circulation, for example - would be helpful.

Finally, perhaps as an alternative to the target 2-based symmetric solution, which may be difficult to implement politically, a reinterpretation of the already existing (but asymmetric) Excessive imbalance prodcedure may be helpful. Alternatively, the target 2-based approach could be integreted into that procedure so that an already existing framwork would be used but just reinterpreted or reinforced.

Author Response

This is a highly topical and relevant paper that should be published. The authors may want to provide some more information as to why their solution is legally and politically feasible in the current EMU framework. Also some technical details and some more information about the risks of the strategy - what happens if too many Geuros come into circulation, for example - would be helpful.

We have extended our discussion to address the referee’s concerns

Finally, perhaps as an alternative to the target 2-based symmetric solution, which may be difficult to implement politically, a reinterpretation of the already existing (but asymmetric) Excessive imbalance prodcedure may be helpful. Alternatively, the target 2-based approach could be integreted into that procedure so that an already existing framwork would be used but just reinterpreted or reinforced.

We have added in Section 4 a discussion of the possible link between Target 2 balances and the imbalances considered in the Excessive imbalance procedure